# nature | methods

**BRIEF COMMUNICATION**

# ColabFold: making protein folding accessible to all

Milot Mirdita [1,10] ✉, Konstantin Schütze [2], Yoshitaka Moriwaki [3,4], Lim Heo [5],
Sergey Ovchinnikov [6,7,10] ✉ and Martin Steinegger [2,8,9,10] ✉

**ColabFold offers accelerated prediction of protein structures and complexes by combining the fast homology search of MMseqs2 with AlphaFold2 or RoseTTAFold. ColabFold's 40−60-fold faster search and optimized model utilization enables prediction of close to 1,000 structures per day on a server with one graphics processing unit. Coupled with Google Colaboratory, ColabFold becomes a free and accessible platform for protein folding. ColabFold is open-source software available at https://github.com/sokrypton/ColabFold and its novel environmental databases are available at https://colab-fold.mmseqs.com.**

Predicting the three-dimensional (3D) structure of a protein from its sequence alone remains an unsolved problem. However, by exploiting the information in multiple sequence alignments (MSAs) of related proteins as the raw input features for end-to-end training, AlphaFold2 (ref. [1]) was able to predict the 3D atomic coordinates of folded protein structures at a median global distance test total score (GDT_TS) of 92.4% in the latest round of the protein folding competition by the international community, CASP14 (Critical Assessment of protein Structure Prediction, round 14) (ref. [2]). The accuracy of many of the predicted structures was within the error margin of experimental structure determination methods. Many ideas of AlphaFold2 were independently reproduced and implemented in RoseTTAFold (ref. [3]). In addition to predictions for single chains, RoseTTAFold and, later, AlphaFold, were also shown to generalize to protein complexes. Evans et al.[4] have since released AlphaFold-multimer, a refined version of AlphaFold2 for the prediction of protein complexes. Thus, two highly accurate open-source prediction methods for single chains and one for protein complexes are now publicly available.

To leverage the power of these methods, researchers require powerful computing capabilities. First, to build diverse MSAs, large collections of protein sequences from public reference[5] and environmental[1,6] databases are searched using the most sensitive homology detection methods, HMMer[7] and HHblits[8], both of which use profile hidden Markov models (HMMs). These environmental databases contain billions of proteins extracted from metagenomic and transcriptomic experiments, which often complement databases dominated by isolated genomes. Due to their large size, searches can take up to hours for a single protein while requiring more than 2 TB of storage space alone. Second, to execute the deep neural networks, graphics processing units (GPUs) with a large amount of GPU RAM (random access memory) are required even for relatively common

protein sizes of ~1,000 residues. For these, however, the MSA generation dominates the overall run time.

To enable researchers without these resources to use AlphaFold2, independent solutions based on Google Colaboratory were developed. Colaboratory is a proprietary version of Jupyter Notebook hosted by Google. It is accessible for free to logged-in users and includes access to powerful GPUs. Concurrently, Tunyasuvunakool et al.[9] developed an AlphaFold2 Jupyter Notebook for Google Colaboratory (referred to as AlphaFold-Colab), for which the input MSA is built by searching with HMMer against the UniProt Reference Clusters (UniRef90) and an eightfold-reduced environmental database. This results in less accurate predictions while still requiring long search times.

Here, we present ColabFold, a fast and easy-to-use software for the prediction of protein structures and homo- and heteromer complexes, for use as a Jupyter Notebook inside Google Colaboratory, on researchers' local computers as a notebook or through a command line interface. ColabFold speeds up single predictions by replacing AlphaFold2's homology search with the 40–60-fold faster MMseqs2 (Many-against-Many sequence searching) (refs. [10,11]), and speeds up batch predictions by ~90-fold by avoiding recompilation and adding an early stop criterion. We show that ColabFold outperforms AlphaFold-Colab and matches AlphaFold2 on CASP14 targets and also matches AlphaFold-multimer on the ClusPro[4,12] dataset in prediction quality.

ColabFold (Fig. 1) consists of three parts. The first is an MMseqs2-based homology search server to build diverse MSAs and to find templates. The server efficiently aligns input sequence(s) against the databanks UniRef100, PDB70 and an environmental sequence set. The second part is a Python library that communicates with the MMseqs2 search server, prepares the input features for structure inference (single chains or complexes), and visualizes the results. This library also implements a command line interface. The last part consists of the Jupyter notebooks for basic, advanced and batch use (Methods 2.1.1) using the Python library.

In ColabFold we replace the sensitive search methods HMMer and HHblits by MMseqs2. We optimized the MSA generation by MMseqs2 to have the following three properties: MSA generation should be fast; the MSA has to capture diversity well; and it has to be small enough to run on computers with limited RAM. Reducing the memory requirement is especially helpful in Google Colaboratory, where the provided system is selected from a pool with widely differing capabilities. While the first requirement is achieved through the fast MMseqs2 prefilter, for the second and third requirements

[1]Quantitative and Computational Biology, Max Planck Institute for Multidisciplinary Sciences, Göttingen, Germany. [2]School of Biological Sciences, Seoul National University, Seoul, South Korea. [3]Department of Biotechnology, Graduate School of Agricultural and Life Sciences, The University of Tokyo, Tokyo, Japan. [4]Collaborative Research Institute for Innovative Microbiology, The University of Tokyo, Tokyo, Japan. [5]Department of Biochemistry and Molecular Biology, Michigan State University, East Lansing, MI, USA. [6]JHDSF Program, Harvard University, Cambridge, MA, USA. [7]FAS Division of Science, Harvard University, Cambridge, MA, USA. [8]Artificial Intelligence Institute, Seoul National University, Seoul, South Korea. [9]Institute of Molecular Biology and Genetics, Seoul National University, Seoul, South Korea. [10]These authors contributed equally: Milot Mirdita, Sergey Ovchinnikov and Martin Steinegger. ✉e-mail: milot.mirdita@mpinat.mpg.de; so@fas.harvard.edu; martin.steinegger@snu.ac.kr

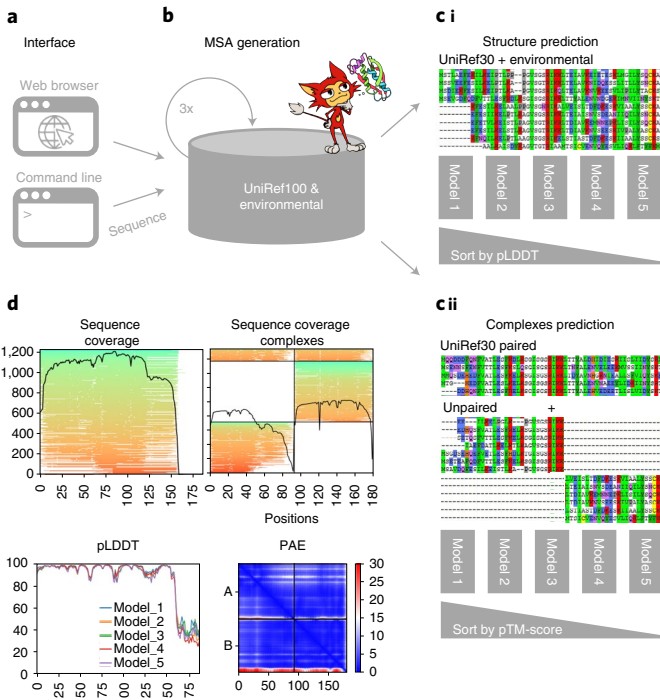

**Fig. 1 | Schematic diagram of ColabFold. a,b,** ColabFold has a web and a command line interface (**a**) that send FASTA input sequence(s) to an MMseqs2 server (**b**) searching two databases, UniRef100 and a database of environmental sequences, with three profile-search iterations each. The second database is searched using a sequence profile generated from the UniRef100 search as input. The server generates two MSAs in A3M format containing all detected sequences. **c,** For predictions of single structures (i) we filter both A3Ms using a diversity-aware filter and return this to be provided as the MSA input feature to the AlphaFold2 models. For predictions of complexes (ii) we pair the top hits within the same species to resolve the inter-chain contacts and additionally add two unpaired MSAs (same as i) to guide the structure prediction. Single chain predictions are ranked by pLDDT and complexes by predicted TM-score. **d,** To help researchers judge the prediction quality we visualize MSA depth and diversity and show the AlphaFold2 confidence measures (pLDDT and PAE).

we developed a search workflow to maximize sensitivity (Methods 2.2.1) and a new filter that samples the sequence space evenly (Methods 2.2.2 and Supplementary Fig. 1). Prediction quality depends on the input MSA, however, often an MSA with only a few (~30) sufficiently diverse sequences is enough to produce high-quality predictions (see fig. 5a in ref. [1]).

Additionally, we combined the Big Fantastic Database (BFD) and the MGnify database, which are used in AlphaFold2 by HHblits and HMMer, respectively, into a combined redundancy-reduced version that we refer to as BFD/MGnify (Methods 2.3.1). The environmental search database presented an opportunity to improve structure predictions of non-bacterial sequences given that, for example, eukaryotic protein diversity is not well represented in the databases BFD and MGnify. Limitations in assembly and gene calling due to complex intron and exon structures result in underrepresentation in reference databases. We therefore extended BFD/MGnify with additional metagenomic protein catalogs containing eukaryotic proteins[13–15], phage catalogs[16,17] and an updated version of MetaClust[18]. We refer to this database as ColabFoldDB (Methods 2.3.2). In Supplementary Fig. 2 we show that ColabFoldDB, in comparison with BFD/MGnify, produces more diverse MSAs for domains in the protein families database (Pfam)[19] with <30 members.

To compare the accuracy of predicted structures we compared AlphaFold2 (default settings with templates), AlphaFold-Colab (no templates), ColabFold-RoseTTAFold-BFD/MGnify, ColabFold-AlphaFold2-BFD/MGnify and ColabFold-AlphaFold2-ColabFoldDB on template modeling scores (TM-scores) for all targets from the CASP14 competition (Fig. 2a). All three ColabFold modes were executed without templates. We show the targets split by free modeling on the left and the remaining ones on the right, given that we used the free-modeling targets for optimization of search workflow parameters. ColabFold is on average fivefold faster for single predictions than AlphaFold2 and AlphaFold-Colab, when taking both MSA generation (Fig. 2b) and model inference into account.

The mean TM-scores for the free-modeling targets are 0.826, 0.818, 0.79, 0.744 and 0.62 for ColabFold-AlphaFold2-BFD/MGnify, ColabFold-AlphaFold2-ColabFoldDB, AlphaFold2, AlphaFold-Colab and ColabFold-RoseTTAFold-BFD/MGnify, respectively. Over all CASP14 targets (excluding AlphaFold-Colab because it cannot be used as a standalone) the TM-scores are 0.887, 0.886, 0.888 and 0.754 for the respective methods. The prediction of target T1084 can be improved from a TM-score of 0.457 to 0.872 by ColabFold if MMseqs2's compositional filter is disabled (Supplementary Fig. 3). Supplementary Table 1 lists the additional targets for which ColabFold differed significantly from AlphaFold2.

AlphaFold2 was initially released without the capability to model protein complexes. However, we found that by combining two sequences with a glycine linker[20] it could often successfully model complexes. Shortly afterwards, Baek[21] found that increasing the model's internal parameter, residue-index (the method that was used in RoseTTAFold), could also be done in AlphaFold2.

For high-quality predictions it was shown that sequences should be provided in paired form to AlphaFold2 (ref. [22]). We implemented a similar pairing procedure (Methods 2.4.2) and show the prediction capabilities of ColabFold for complexes in Fig. 2c. ColabFold achieves the highest accuracy in the prediction of complexes on the ClusPro[4,12] dataset with the AlphaFold-multimer model, however, some targets performed better using the residue-index mode.

Supplementary Fig. 4a,b show two examples of ColabFold's prediction capabilities for complexes. Supplementary Fig. 4a shows a homo-six-mer and Supplementary Fig. 4b shows a D-methionine transport system composed of three different proteins. The inter-chain predicted alignment error (inter-PAE) provided by AlphaFold2 helps to rank the complexes. Plots of PAE and complex conformation are given to help users judge the prediction quality of a complex. An example for heteromer complex prediction is shown in Supplementary Fig. 4c with its PAE plot. ColabFold complexes were successfully used to aid in the determination of the structure of the 120 MDa human nucleopore complex on cryogenic electron microscopy[23].

ColabFold exposes many internal parameters of AlphaFold2 such as the recycle count (default 3), which controls the number of times the prediction is repeatedly fed through the model. For difficult targets as well as for designed proteins without known homologs, additional recycling iterations can result in a high-quality prediction (Supplementary Fig. 5). Rerunning the CASP14 benchmark with a recycle count of 12 resulted in an improvement of targets with little MSA information, resulting in an increased average TM-score of 0.898 (Supplementary Fig. 6).

For high-throughput structure prediction, we introduced several features in ColabFold. First, MSA generation can be executed in batch mode independently from model batch-inference. Second, we compile only one of the five AlphaFold2 models and reuse weights. Third, we avoid recompilation for sequences of similar length. Fourth, we implement early stop criteria, to avoid additional recycles or models if a sufficiently accurate structure was already found. And last, we developed the command line tool `colabfold_batch` to

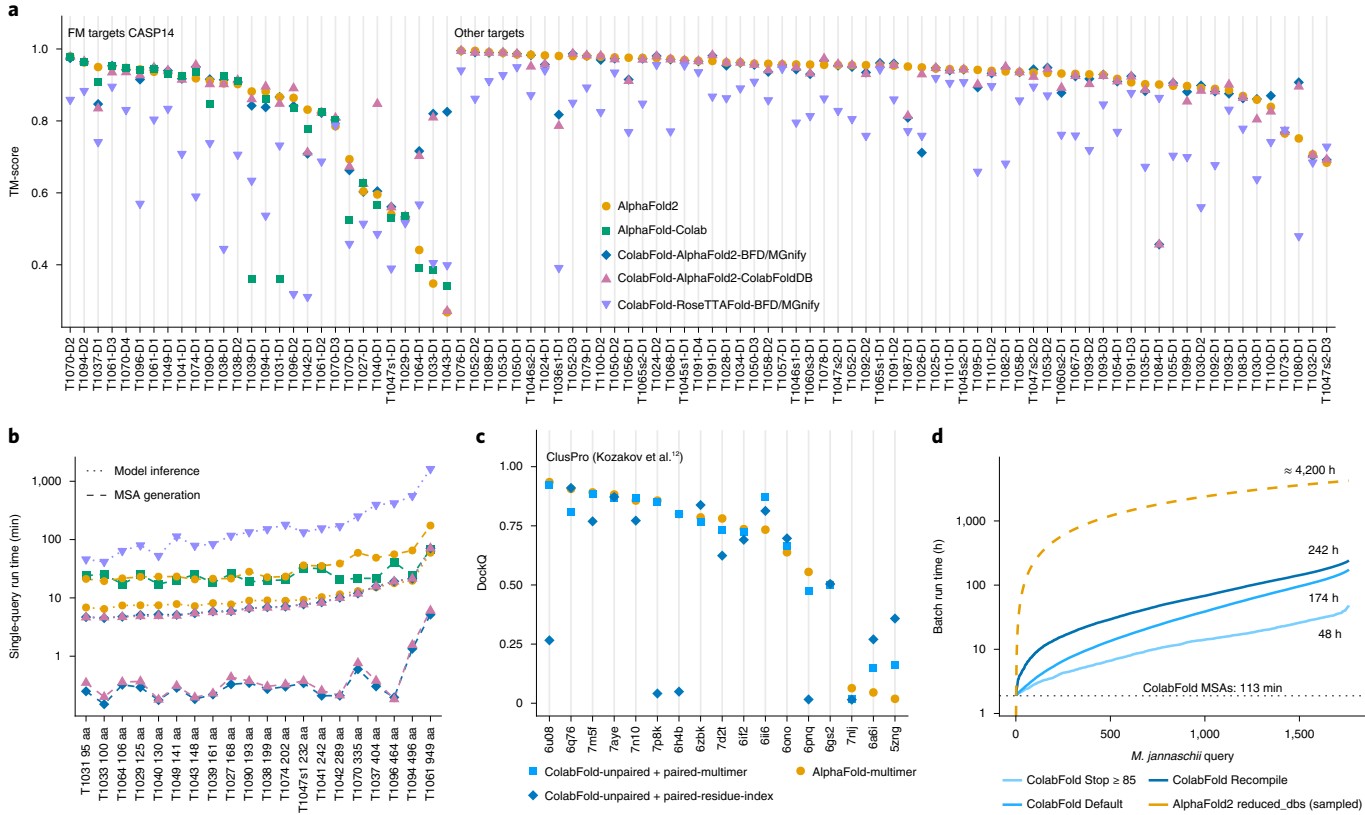

**Fig. 2 | Comparison of predictions for single chains and complexes. a,** Structure prediction comparison of AlphaFold2, AlphaFold-Colab and ColabFold-AlphaFold2 with BFD/MGnify and with the ColabFoldDB, and ColabFold-RoseTTAFold with BFD/MGnify using predictions of 91 domains of 65 CASP14 targets. The 28 domains from the 20 free-modeling (FM) targets are shown first. FM targets were used to optimize MMseqs2 search parameters. Each target was evaluated for each individual domain (in total 91 domains). **b,** MSA generation and model inference times for each CASP14 FM target sorted by protein length (same colors as before). Blue shows MSA run times for ColabFold-AlphaFold2-BFD/MGnify and ColabFold-RoseTTAFold-BFD/MGnify. **c,** Comparison of multimeric prediction modes in ColabFold and AlphaFold-multimer. The ColabFold modes include residue-index modification with models originally trained for single-chain predictions and those for multimeric prediction from AlphaFold-multimer, using DockQ (a quality measure for protein–protein docking models). **d,** Run time of colabfold_batch proteome prediction at three optimization levels: always recompile, default, and stop model/recycle evaluation after first prediction with a pLDDT of ≥85. The yellow dashed line represents an extrapolation on the basis of the 50 AlphaFold2 predictions.

predict structures on local machines. All together, we show that the *Methanocaldococcus jannaschii* proteome of 1,762 proteins shorter than 1,000 amino acids can be predicted in 48 h with early stopping at a pLDDT (predicted local distance difference test; a per-residue confidence metric) of ≥85 on one Nvidia Titan RTX (Fig. 2d), while sacrificing little or no prediction accuracy (Methods 2.7.4). The average pLDDTs of AlphaFold2 and ColabFold Stop ≥ 85 were 89.75 and 88.78 in a subsampled set of 50 proteins.

ColabFold builds beyond the initial offerings of Alphafold2 by improving its sequence search, providing tools for modeling homo- and heteromer complexes, exposing advanced functionality, expanding the environmental databases and enabling large-scale batch prediction of protein structures, at an approximately 90-fold speed-up over AlphaFold2.

## Online content

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

## Methods

**Executing ColabFold.** ColabFold is available as a set of Jupyter notebooks that can be used on Google Colaboratory or on users' local machines, as well as an easily installable command line application.

*ColabFold notebooks.* ColabFold has four main Jupyter notebooks[24]. The first is AlphaFold2_mmseqs2 for basic use, which supports protein structure prediction using MSAs generated by MMseqs2 (version edb822), custom MSA upload, use of template information, relaxing of the predicted structures using amber force fields[25], and prediction of complexes. The second, AlphaFold2_advanced, for advanced users, additionally supports MSA generation using HMMer (same as AlphaFold-Colab), the sampling of diverse structures by iterating through a series of random seeds (num_samples), and control of AlphaFold2 model internal parameters, such as changing the number of recycles (max_recycle), the number of ensembles (num_ensemble), and the is_training option. The use of the is_training option enables dropout during inference. This activates the stochastic part of the model and can result in different predictions. Thus by iterating through different seeds, one can sample different structures predictions from the uncertainty of the model[26] or the ambiguity of co-evolution constraints derived from the input MSA. The third main type of Jupyter notebook is AlphaFold2_batch, for batch prediction of multiple sequences or MSAs. The batch notebook saves time by avoiding recompilation of the AlphaFold2 models (section 2.5.2) for each individual input sequence. The fourth type is RoseTTAFold, for basic use of RoseTTAFold, and which supports protein structure prediction using MSAs generated by MMseqs2, and custom MSAs, and sidechain prediction using SCWRL4 (ref. [27]). The RoseTTAFold notebook also has an option to use a slower but more accurate PyRosetta[28] folding protocol for structure prediction, using constraints predicted by RoseTTAFold's neural network.

*ColabFold command line interface.* We initially focused on making ColabFold as widely available as possible through our Notebooks running in Google Colaboratory. To meet the demand for a version that runs on local users' machines, we released 'LocalColabFold'. LocalColabFold can take command line arguments to specify an input FASTA file, an output directory, and various options to tweak structure predictions. LocalColabFold runs on a wide range of operating systems, such as Windows 10 or later (using Windows Subsystem for Linux 2), macOS and Linux. The structure inference and energy minimization are accelerated if a CUDA 11.1 or later compatible GPU is present. LocalColabFold is available as free open-source software at https://github.com/YoshitakaMo/localcolabfold

Recognizing the limitations of Google Colaboratory, we provide the colabfold_batch command line tool through the colabfold python package. This enables the computing of tasks on the user's own computer that would have been too large for Google Colab, for example, predicting an entire proteome (Methods 2.7.4). It can be installed following Python's pip package manager following the instructions at https://github.com/sokrypton/ColabFold. It can be used as colabfold_batch input_file_or_directory output_directory, supporting FASTA, A3M and CSV files as input.

**Faster MSA generation with MMseqs2.** Generating MSAs for AlphaFold2 and RoseTTAFold is a time-consuming task. To improve their run time while maintaining a high prediction accuracy, we implemented optimized workflows using MMseqs2.

*MSA generation by MMseqs2.* ColabFold sends the query sequence to an MMseqs2 server[11]. It searches the sequence(s) with three iterations against the consensus sequences of the UniRef30, a clustered version of the UniRef100 (ref. [29]). We accept hits with an E-value lower than 0.1. For each hit we realign its respective UniRef100 cluster member using the profile generated by the last iterative search, filter them (Methods 2.2.2) and add these to the MSA. This expanding search results in a speed-up of ~10-fold given that only 29.3 million cluster consensus sequences are searched instead of all 277.5 million UniRef100 sequences. Additionally, it has the advantage of being more sensitive given that the cluster consensus sequences are used. We use the UniRef30 sequence profile to perform an iterative search against the BFD/MGnify or ColabFoldDB using the same parameters, filters and expansion strategy.

*New diversity aware filter.* To limit the number of hits in the final MSA we use the HHblits (v3.3.0) diversity filtering algorithm[8] implemented in MMseqs2 in multiple stages. In the first stage, during UniRef cluster expansion, we filter each individual UniRef30 cluster before adding the cluster members to the MSA, such that no cluster pair has a higher maximum sequence identity than 95% (--max-seq-id 0.95). In the second stage, after realignment we enable only the --qsc 0.8 threshold and disable all other thresholds (--qid 0 --diff 0 --max-seq-id 1.0). Additionally, the qsc filtering is used only if at least 100 hits are found (--filter-min-enable 100). In the last stage, during MSA construction we filter again with the following parameters: --filter-min-enable 1000 --diff 3000 --qid 0.0,0.2,0.4,0.6,0.8,1.0 --qsc 0 --max-seq-id 0.95. Here, we extended the HHblits filtering algorithm to filter within a given sequence identity bucket such that it cannot eliminate redundancy across filter buckets. Our

filter keeps the 3,000 most diverse sequences in the identity buckets [0.0–0.2], (0.2–0.4], (0.4–0.6], (0.6–0.8] and (0.8–1.0]. In buckets containing less than 1,000 hits we disable the filtering.

*New MMseqs2 pre-computed index to support expanding cluster members.* MMseqs2 was initially built to perform fast many-against-many sequence searches. Mirdita et al.[11] improved it to also support fast single-against-many searches. This type of search requires the database to be indexed and stored in memory. mmseqs createindex indexes the sequences and stores all time-consuming-to-compute data structures used for MMseqs2 searches to disk. We load the index into the operating systems cache using vmtouch (https://github.com/hoytech/vmtouch) to enable calls to the different MMseqs2 modules to become nearly overhead free. We extended the index to store, in addition to the already present cluster consensus sequences, all member sequences and the pairwise alignments of the cluster representatives to the cluster members. With these resident in cache, we eliminate the overhead of the remaining module calls.

**ColabFold databases.** AlphaFold2 requires more than 2 TB of storage space for its databases, which is a significant hurdle for many researchers. We optimized its databases and additionally created another large environmental sequence database.

*Reducing the size of BFD/MGnify.* To keep all required sequences and data structures in memory we needed to reduce the size of the environmental databases BFD and MGnify, given that both databases together would have required ~517 GB RAM for headers and sequences alone.

BFD is a clustered protein database consisting of ~2.2 billion proteins organized in 64 million clusters. MGnify (2019_05) contains ~300 million environmental proteins. We merged both databases by searching the MGnify sequences against the BFD cluster-representative sequences using MMseqs2. Each MGnify sequence with a sequence identity of >30% and a local alignment that covers at least 90% of its length is assigned to the respective BFD cluster. All unassigned sequences are clustered at 30% sequence identity and 90% coverage (--min-seq-id 0.3 -c 0.3 --cov-mode 1 -s 3) and merged with the BFD clusters, resulting in 182 million clusters. To reduce the size of the database we filtered each cluster, keeping only the 10 most diverse sequences using mmseqs filterresult --diff 10. This reduced the total number of sequences from 2.5 billion to 513 million, thus requiring only 84 GB RAM for headers and sequences.

*ColabFoldDB.* We built ColabFoldDB by expanding BFD/MGnify with metagenomic sequences from various environments. To update the database we searched the proteins from the SMAG (eukaryotes)[14], MetaEuk (eukaryotes)[13], TOPAZ (eukaryotes)[15], MGV (DNA viruses)[16], GPD (bacteriophages)[17] and an updated version of MetaClust[18] against the BFD/MGnify centroids using MMseqs2 and assigned each sequence to the respective cluster if they have a 30% sequence identity at a 90% sequence overlap (-c 0.9 --cov-mode 1 --min-seq-id 0.3). All remaining sequences were clustered using MMseqs2 cluster -c 0.9 --cov-mode 1 --min-seq-id 0.3 and appended to the database. We remove redundancy per cluster by keeping the most 10 diverse sequences using mmseqs filterresult --diff 10. The final database consists of 209,335,865 million representative sequences and 738,695,580 members (see the Data Availability section for the input files). We provide the MMseqs2 search workflow used in the server (Methods 2.2.1) as a standalone script (colabfold_search).

*Template information.* AlphaFold2 searches with HHsearch through a clustered version of the PDB (PDB70, ref. [8]) to find the 20 top ranked templates. To save time, we use MMseqs2 (ref. [10]) to search against the PDB70 cluster representatives as a prefiltering step to find candidate templates. This search is also done as part of the MMseqs2 API call on our server. Only the top 20 target templates according to E-value are then aligned by HHsearch. The accepted templates are given to AlphaFold2 as input features. This alignment step is done in the ColabFold client and therefore it requires the subset of the PDB70 containing the respective HMMs. The PDB70 subset and the PDB mmCIF files are fetched from our server. For benchmarking, no templates are given to ColabFold.

**Modeling protein complexes with ColabFold.** ColabFold offers protein complex folding through the specialized AlphaFold-multimer model and through manipulation of the residue index[3]. Here, we show the steps that we took for ColabFold to produce accurate protein complex predictions.

*Modeling of protein–protein complexes.* We implemented two protein complex prediction modes in ColabFold. One is based on AlphaFold-multimer[4] and the other is based on the manipulation of residue index in the original AlphaFold2 model. Baek et al.[3] show that RoseTTAFold is able to model complexes despite being trained only on single chains. This is done by providing a paired alignment and modifying the residue index. The residue index is used as an input to the models to compute positional embedding. In AlphaFold2 we find the same to be true, although surprisingly the paired alignment is often not needed (Fig. 2c). AlphaFold2 uses relative positional encoding with a cap at $|i - j| \geq 32$, meaning that any pair of residues separated by 32 or more are given the same relative

positional encoding. By offsetting the residue index between two proteins to be > 32, AlphaFold2 treats them as separate polypeptide chains. ColabFold integrates this for modeling complexes.

For homo-oligomeric complexes (Supplementary Fig. 4a) the MSA is copied multiple times for each component. Interestingly, it was found that providing a separate MSA copy (padding by gap characters to extend to other copies) works significantly better than concatenating from left to right.

For hetero-oligomeric complexes (Supplementary Fig. 4b), a separate MSA is generated for each component. The MSA is paired according to the chosen `pair_mode` (section 2.4.2). Given that pLDDT is useful only for assessing local structure confidence, we use the fine-tuned model parameters to return the PAE for each prediction. As illustrated in Supplementary Fig. 4c, the inter-PAE, the predicted TM-score or interface TM-score (both derived from PAE) can be used to rank and assess the confidence of the predicted protein–protein interaction.

*MSA pairing for complex prediction.* A paired MSA helps AlphaFold2 to predict complexes more accurately only if orthologous genes are paired with each other. We followed a similar strategy as Bryant et al.[22] to pair sequences according to their taxonomic identifier. For the pairing we search each distinct sequence of a complex against the UniRef100 using the same procedure as described in section 2.2.1. We return only hits that cover all complex proteins within one species and pair only the best hit (smallest E-value) with an alignment that covers the query to at least 50%. The pairing is implemented in the new MMseqs2 module `pairaln`.

For prokaryotic protein prediction we additionally implemented the protocol described in ref. [3] to pair sequences based on their distances in the genome as predicted from the UniProt accession numbers.

*Taxonomic labels for MSA pairing.* To pair MSAs for complex prediction, we retrieve for each found UniRef100 member sequence the taxonomic identifier from the NCBI (National Center for Biotechnology Information) Taxonomy database[30]. The taxonomic labels are extracted from the lowest common ancestor field ('common taxon ID') of each UniRef100 sequence from the `uniref100.xml` (2021_03) file.

**Speeding up AlphaFold2's model evaluation.** Our efforts in speeding up AlphaFold2's MSA generation yielded large improvements in its run time. However, we discovered multiple opportunities within AlphaFold2 to speed up its model inference without sacrificing (or only sacrificing very little of) its accuracy.

*Avoid recompiling AlphaFold2 models.* The AlphaFold2 models are compiled using JAX[31] to optimize the model for specific MSA or template input sizes. When no templates are provided, we compile once and, during inference, replace the weights from the other models, using the configuration of model 5. This saves 7 min of compile time. When templates are enabled, model 1 is compiled and weights from model 2 are used, model 3 is compiled and weights from models 4 and 5 are used. This saves 5 min of compile time. If the user changes the sequence or settings without changing the length or number of sequences in the MSA, the compiled models are reused without triggering recompilation.

*Avoid recompiling during batch computation.* To avoid AlphaFold2 model recompilation for every protein AlphaFold2 provides a function to add padding to the input MSA and templates called `make_fixed_size`. However, this is not exposed in AlphaFold2. We used the function in our batch notebook as well as in our command line tool colabfold_batch, to maximize GPU use and minimize the need for model recompilation. We sort the input queries by sequence length and process them in ascending order. We pad the input features by 10% (by default). All sequences that lie within the query length and an additional 10% margin are not required to be recompiled, resulting in a large speed-up for short proteins.

*Recycle count.* AlphaFold2 improves the predicted protein structure by recycling (by default) three times, meaning that the prediction is fed multiple times through the model. We exposed the recycle count as a customizable parameter given that additional recycles can often improve a model (Supplementary Fig. 6) at the cost of a longer run time. We also implemented an option to specify a tolerance threshold to stop early. For some designed proteins without known homologous sequences, this helped to fold the final protein (Supplementary Fig. 5).

*Speed-up of predictions through early stop.* AlphaFold2 computes five models through multiple recycles. We noted that for prediction of high certainty (pLDDT >85), all five models would often produce structures of very similar confidence, for some even without or with less than three recycles. To speed up the computation we added a parameter to define an early stop criterion that halts additional model inferences and stops recycling if a given pLDDT or (interface) predicted TM-score threshold is reached.

**Advanced features.** In our investigation of AlphaFold2's internal parameters we realized that we could expose many of the internal parameters that might be useful to researchers trying to explore AlphaFold2's full potential.

*Sampling of diverse structures.* To reduce memory requirements, only a subset of the MSA is used as input to the model. AlphaFold2, depending on model configuration, subsamples the MSA to a maximum of 512 cluster centers and 1,024 extra sequences. Changing the random seed can result in different cluster centers and thus different structure predictions. ColabFold provides an option to iterate through a series of random seeds, resulting in structure diversity. Further structure diversity can be generated by using the original or fine-tuned (`use_ptm`) model parameters and/or enabling `is_training` to activate the stochastic (dropout) part of the model. Enabling the latter can be used to sample an ensemble of models for the uncertain parts of the structure prediction.

*Custom MSAs.* ColabFold enables researchers to upload their own MSAs. Any kind of alignment tool can be used to generate the MSA. The uploaded MSA can be provided in aligned FASTA, A3M, STOCKHOLM or Clustal format. We convert the respective MSA format into A3M format using the `reformat.pl` script from the HH-suite[8].

*Lightweight 2D structure renderer.* For visualization, we developed a matplotlib[32] compatible module for displaying the 3D ribbon diagram of a protein structure or complex. The ribbon can be colored by residue index (N to C terminus) or by a predicted confidence metric (such as pLDDT). For complexes, each protein can be colored by chain ID. Instead of using a 3D renderer, we instead use a 2D line plotting based technique. The lines that make up the ribbon are plotted in the order in which they appear along the z-axis. Furthermore, we add shade to the lines according to the z-axis. This creates the illusion of a 3D rendered graphic. The advantage over a 3D renderer is that the images are very lightweight, can be used in animations and saved as vector graphics for lossless inclusion in documents. Given that the 2D renderer is not interactive, we additionally included a 3D visualization option using py3Dmol[33] in the ColabFold notebooks.

**Benchmarking ColabFold.** We show with multiple datasets that ColabFold does not sacrifice accuracy for its much faster run times.

*Benchmark with CASP14 targets.* We compared AlphaFold-Colab and AlphaFold2 (commit `b88f8da`) against ColabFold using all CASP14 (ref. [2]) targets. ColabFold-AlphaFold2 (commit `2b49880`) used UniRef30 (2021_03) (ref. [34]) and the BFD/MGnify or ColabFoldDB. ColabFold-RoseTTAFold (commit `ae2b519`) was executed with papermill (https://github.com/nteract/papermill) using the PyRosetta protocol[28]. ColabFold-RoseTTAFold-BFD/MGnify and ColabFold-AlphaFold2-BFD/MGnify used the same MSAs. AlphaFold-Colab used the UniRef90 (2021_03), MGnify (2019_05) and the small BFD. AlphaFold2 used the `full_dbs` preset and default databases downloaded with the `download_all_data.sh` script. The 65 targets contain 91 domains, among these are 20 free-modeling targets with 28 domains. We compared the predictions against the experimental structures using TMalign (downloaded on 24 February 2021) (ref. [35]).

*Measuring run times for CASP14 benchmark.* To provide more accurate run times we split the MSA generation and model inference measurements. MSA generation was repeated five times and the MSA generation times were averaged.

ColabFold was executed using `colabfold_batch`. The MMseqs2 server that computes MSAs for ColabFold has 2 × 14 core Intel E5-2680v4 central processing units (CPUs) and 768 GB RAM. Each generated MSA was processed by a single CPU core. Run times were computed from server logs.

AlphaFold2 MSA generation run times were measured by running AlphaFold2 without models (providing an empty string to the `--model_names` parameter) on the same 2 × 14 core Intel E5-2680v4 CPUs and 768 GB RAM system. The AlphaFold2 databases were stored on a software-RAID5 as implemented in Linux (mdadm) composed of six Samsung 970 EVO Plus 1 TB NVMe drives. Run times for AlphaFold2 were taken from the `features` entry of the `timings.json` file. For a fair comparison, AlphaFold2 was modified to allow HMMer and HHblits to access one CPU core.

All ColabFold and AlphaFold2 model inference run-time measurements were done on systems with 2 × 16 core Intel Gold 6242 CPUs with 192 GB RAM and 4x Nvidia Quadro RTX5000 GPUs. Only one GPU was used in each run.

ColabFold-RoseTTAFold-BFD/MGnify and ColabFold-AlphaFold2-BFD/MGnify used the same MSAs, and run times are shown only once.

AlphaFold-Colab was executed in the browser using a Google Colab Pro account. The times for the homology search were taken from the notebook output cell 'Search against genetic databases'. The JackHMMer search uses eight threads.

*Complex benchmark.* We compare predictions of 17 ClusPro[4,12] targets to their native structures using DockQ (commit 3735c16) (ref. [36]). We used `colabfold_batch` (commit 45ad0e9) with BFD/MGnify in residue index manipulation and AlphaFold-multimer mode to predict structures. We use MSA pairing as described in section 2.4.2 and also add unpaired sequences. Models are ranked by predicted interface TM-score as returned by AlphaFold-multimer. The DockQ AlphaFold-multimer reference numbers were provided by R. Evans.

*Proteome benchmark.* We predict the proteome of *M. jannaschii*. Of the 1,787 proteins, we exclude the 25 proteins longer than 1,000 residues, leaving 1,762 proteins of 268 amino acids in average length. With the `colabfold_search` wrapper to MMseqs2 we search against the ColabFoldDB (section 2.3.2) in 113 min on a system with an AMD EPYC 7402P 24-core CPU (no hyperthreading) and 512 GB RAM. MMseqs2 had a maximum resident set size of 308 GB during the search. We then predict the structures on a single Nvidia Titan RTX with 24 GB RAM in 46 h using only MSAs (no templates). For each query we stop early if any recycle iteration reaches a pLDDT of at least 85. Early stopping results in a speed-up of 3.7-fold compared with the default and 4.8-fold compared with always recompiling. AlphaFold2 (reduced_dbs) was run with the reduced_dbs preset and no template information was used. We changed the AlphaFold2 source code to utilize all CPU cores during the homology search.

AlphaFold2 (reduced_dbs, v2.1.1), ColabFold (commit f5d0cec) default and ColabFold Stop ≥ 85 have an average pLDDT of 90.68, 90.22 and 89.33, respectively, for 50 randomly sampled proteins. These are the same proteins that were used to extrapolate the run time of AlphaFold2. Over all predictions, the pLDDTs for the *M. jannaschii* proteome downloaded from the AlphaFoldDB, ColabFold default and ColabFold Stop ≥ 85 are 89.75, 89.38 and 88.77, respectively.

*Software used for analysis.* Benchmark data analysis and visualization were done with R/4.1.1, ggplot2/3.3.5, cowplot/1.1.1 and lubridate/1.7.10. ColabFold-generated plots were made using matplotlib/3.1.3. TM-score analysis was done with TMalign/2021/02/24 and DockQ/3735c16.

**Reporting summary.** Further information on research design is available in the Nature Research Reporting Summary linked to this article.

## Data availability

ColabFold databases are freely (CC-BY-SA 4.0) available at https://colabfold.mmseqs.com. MSAs and structures produced during benchmarking: https://wwwuser.gwdg.de/~compbiol/colabfold/manuscript. Input databases used for building ColabFold databases: UniRef30, https://uniclust.mmseqs.com; BFD, https://bfd.mmseqs.com; MGnify, http://ftp.ebi.ac.uk/pub/databases/metagenomics/peptide_database/2019_05; PDB70, https://wwwuser.gwdg.de/~compbiol/data/hhsuite/databases/hhsuite_dbs; MetaEuk, https://wwwuser.gwdg.de/~compbiol/metaeuk/2019_11/MetaEuk_preds_Tara_vs_euk_profiles_uniqs.fas.gz; SMAG, https://www.genoscope.cns.fr/tara/localdata/data/SMAGs-v1/SMAGs_v1_concat.faa.tar.gz; TOPAZ, https://osf.io/gm564; MGV, https://portal.nersc.gov/MGV/MGV_v1.0_2021_07_08/mgv_proteins.faa; and GPD, http://ftp.ebi.ac.uk/pub/databases/metagenomics/genome_sets/gut_phage_database/GPD_proteome.faa.gz. Further datasets used for benchmarking ColabFold: PFAM (Pfam-A.seed.gz and Pfam-A.full.gz), http://ftp.ebi.ac.uk/pub/databases/Pfam/releases/Pfam34.0; and *M. jannaschii* proteome, https://uniprot.org/proteomes/UP000000805 and https://ftp.ebi.ac.uk/pub/databases/alphafold/v1/UP000000805_243232_METJA_v1.tar. Source data are provided with this paper.

## Code availability

ColabFold is free open-source software (MIT) and available at https://github.com/sokrypton/ColabFold. A locally installable version is available at https://github.com/YoshitakaMo/localcolabfold. The ColabFold development version shown in this manuscript is available at https://github.com/konstin/ColabFold. The ColabFold server components are free open-source software (GPLv3) and are available at https://github.com/soedinglab/mmseqs2-app. MMseqs2 is free open-source software (GPLv3) and is available at https://mmseqs.com.

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

## Acknowledgements

The authors thank J. Söding for providing computational resources; R. Evans, J. Jumper and T. Green for answering questions regarding AF2; M. Baek for the complex residue trick; D.-Y. Kim for creating the ColabFold logo; P. Wang, J. Kosinski, S. Cha, E. Guerrero-Araya, K. Dalton, K. Pan, Y. Eliaz, J. Kaczmarzyk, A. Ljubetič and J. Bard for contributions and bug fixes; H. Onoda, K. Yamamoto, R. Kojima, Y. Peng and M. Ohue for contributions and bug fixes to localcolabfold; A. Markovich and J. Varga for notifying us about MSA quality issues; and H. Alexander for providing the TOPAZ proteins as a single file to download. The authors also thank all users for using ColabFold and reporting issues. This work used the Scientific Compute Cluster at GWDG, the joint data center of the Max Planck Society for the Advancement of Science (MPG) and University of Göttingen. M.M. acknowledges the BMBF CompLifeSci project horizontal4meta. M.S. acknowledges support from the National Research Foundation of Korea grant [2019R1A6A1A10073437, 2020M3A9G7103933, 2021R1C1C102065, 2021M3A9I4021220]; New Faculty Startup Fund and the Creative-Pioneering Researchers Program through Seoul National University. Y.M. acknowledges support from Platform Project for Supporting Drug Discovery and Life Science Research (Basis for Supporting Innovative Drug Discovery and Life Science Research (BINDS)) from AMED under grant number JP21am0101107. S.O. is supported by NIH grants DP5OD026389 and R21AI156595, NSF grant MCB2032259 and the Moore-Simons Project on the Origin of the Eukaryotic Cell, Simons Foundation 735929LPI (https://doi.org/10.46714/735929LPI). Any opinions, findings and conclusions or recommendations expressed in this material are those of the author(s) and do not necessarily reflect the views of the National Science Foundation.

## Author contributions

M.M., K.S., S.O. and M.S. performed research and programming, M.M., S.O. and M.S. jointly designed the research and wrote the manuscript. Y.M. provided the initial methodology for hetero-complex modeling and created an installer for use on local servers. L.H. provided the initial benchmarking.

## Funding

## Competing interests

The authors declare no competing interests.

## Additional information

**Correspondence and requests for materials** should be addressed to Milot Mirdita, Sergey Ovchinnikov or Martin Steinegger.

# nature research

# Reporting Summary

Nature Research wishes to improve the reproducibility of the work that we publish. This form provides structure for consistency and transparency in reporting. For further information on Nature Research policies, see our Editorial Policies and the Editorial Policy Checklist.

## Statistics

For all statistical analyses, confirm that the following items are present in the figure legend, table legend, main text, or Methods section.

| n/a | Confirmed | |
|---|---|---|
| ☐ | ☒ | The exact sample size (*n*) for each experimental group/condition, given as a discrete number and unit of measurement |
| ☒ | ☐ | A statement on whether measurements were taken from distinct samples or whether the same sample was measured repeatedly |
| ☒ | ☐ | The statistical test(s) used AND whether they are one- or two-sided *Only common tests should be described solely by name; describe more complex techniques in the Methods section.* |
| ☒ | ☐ | A description of all covariates tested |
| ☒ | ☐ | A description of any assumptions or corrections, such as tests of normality and adjustment for multiple comparisons |
| ☒ | ☐ | A full description of the statistical parameters including central tendency (e.g. means) or other basic estimates (e.g. regression coefficient) AND variation (e.g. standard deviation) or associated estimates of uncertainty (e.g. confidence intervals) |
| ☒ | ☐ | For null hypothesis testing, the test statistic (e.g. *F*, *t*, *r*) with confidence intervals, effect sizes, degrees of freedom and *P* value noted *Give P values as exact values whenever suitable.* |
| ☒ | ☐ | For Bayesian analysis, information on the choice of priors and Markov chain Monte Carlo settings |
| ☒ | ☐ | For hierarchical and complex designs, identification of the appropriate level for tests and full reporting of outcomes |
| ☒ | ☐ | Estimates of effect sizes (e.g. Cohen's *d*, Pearson's *r*), indicating how they were calculated |

*Our web collection on statistics for biologists contains articles on many of the points above.*

## Software and code

Policy information about availability of computer code

| Data collection | ColabFold is free open-source software (MIT) and available at https://github.com/sokrypton/ColabFold. A locally installable version (MIT) is available at https://github.com/YoshitakaMo/localcolabfold. The ColabFold development version shown in this manuscript is available at https://github.com/konstin/ColabFold. This version will shortly be integrated into the main repository. The ColabFold server components are free open-source software (GPLv3) and available at https://github.com/soedinglab/mmseqs2-app. MMseqs2 is free open-source software (GPLv3) and available at https://mmseqs.com. The ColabFold databases are available at https://colabfold.mmseqs.com under CC-BY 4.0 license. |
|---|---|
| Data analysis | Benchmark data analysis and visualization was done with R/4.1.1, ggplot/3.3.5, cowplot/1.1.1, lubridate/1.7.10. ColabFold generated plots were made using matplotlib/3.1.3. TM-score analysis was done with TMalign/2021/02/24 and DockQ/3735c16. Data was generated with the following software: MMseqs2 (github commit edb822), Colabfold (github commit 45ad0e9), RoseTTAFold (github commit fcf9125), HHblits v3.3.0 and AlphaFold2 v2.1.1 |

For manuscripts utilizing custom algorithms or software that are central to the research but not yet described in published literature, software must be made available to editors and reviewers. We strongly encourage code deposition in a community repository (e.g. GitHub). See the Nature Research guidelines for submitting code & software for further information.

## Data

Policy information about [availability of data](availability of data)

All manuscripts must include a [data availability statement](data availability statement). This statement should provide the following information, where applicable:
- Accession codes, unique identifiers, or web links for publicly available datasets
- A list of figures that have associated raw data
- A description of any restrictions on data availability

Code Availability
ColabFold is free open-source software (MIT) and available at github.com/sokrypton/ColabFold.
A locally installable version is available at github.com/YoshitakaMo/localcolabfold.
The ColabFold development version shown in this manuscript is available at github.com/konstin/ColabFold.
The ColabFold server components are free open-source software (GPLv3) and available at github.com/soedinglab/mmseqs2-app.
MMseqs2 is free open-source software (GPLv3) and available at mmseqs.com.

Data Availability
ColabFold databases are freely (CC-BY-SA 4.0) available at colabfold.mmseqs.com.
MSAs and structures produced during benchmarking:
wwwuser.gwdg.de/~compbiol/colabfold/manuscript
Input databases used for building ColabFold databases:
UniRef30: uniclust.mmseqs.com
BFD: bfd.mmseqs.com
MGnify:ftp.ebi.ac.uk/pub/databases/metagenomics/peptide_database/2019_05
PDB70: wwwuser.gwdg.de/~compbiol/data/hhsuite/databases/hhsuite_dbs
MetaEuk: wwwuser.gwdg.de/~compbiol/metaeuk/2019_11/MetaEuk_preds_Tara_vs_euk_profiles_uniqs.fas.gz
SMAG: www.genoscope.cns.fr/tara/localdata/data/SMAGs-v1/SMAGs_v1_concat.faa.tar.gz
TOPAZ: osf.io/gm564
MGV: portal.nersc.gov/MGV/MGV_v1.0_2021_07_08/mgv_proteins.faa
GPD: ftp.ebi.ac.uk/pub/databases/metagenomics/genome_sets/gut_phage_database/GPD_proteome.faa
Further datasets used for benchmarking ColabFold:
PFAM (Pfam-A.seed.gz & Pfam-A.full.gz): ftp.ebi.ac.uk/pub/databases/Pfam/releases/Pfam34.0
textit{M. jannaschii proteome: uniprot.org/proteomes/UP000000805 ftp.ebi.ac.uk/pub/databases/alphafold/v1/UP000000805_243232_METJA_v1.tar

# Field-specific reporting

Please select the one below that is the best fit for your research. If you are not sure, read the appropriate sections before making your selection.

☒ Life sciences          ☐ Behavioural & social sciences          ☐ Ecological, evolutionary & environmental sciences

For a reference copy of the document with all sections, see [nature.com/documents/nr-reporting-summary-flat.pdf](nature.com/documents/nr-reporting-summary-flat.pdf)

# Life sciences study design

All studies must disclose on these points even when the disclosure is negative.

| | |
|---|---|
| Sample size | ColabFold was evaluated on all CASP14 targets for single-chain predictions. For complex predictions, ColabFold was evaluated on the publicly available ClusPro dataset. We do not compute sample size since previously published standard benchmark sets are used. |
| Data exclusions | No targets were excluded. |
| Replication | Not applicable. ColabFold is exclusively a computational method. The computional method is deterministic (same result each time you run) when run on the same computer setup. This is why replicates are not needed, as the result would be identical for each replicate. |
| Randomization | Not applicable. We are not comparing across groups. |
| Blinding | Not applicable. We are not comparing across groups. |

# Reporting for specific materials, systems and methods

We require information from authors about some types of materials, experimental systems and methods used in many studies. Here, indicate whether each material, system or method listed is relevant to your study. If you are not sure if a list item applies to your research, read the appropriate section before selecting a response.

## Materials & experimental systems

| n/a | Involved in the study |
|-----|----------------------|
| ☒ | ☐ Antibodies |
| ☒ | ☐ Eukaryotic cell lines |
| ☒ | ☐ Palaeontology and archaeology |
| ☒ | ☐ Animals and other organisms |
| ☒ | ☐ Human research participants |
| ☒ | ☐ Clinical data |
| ☒ | ☐ Dual use research of concern |

## Methods

| n/a | Involved in the study |
|-----|----------------------|
| ☒ | ☐ ChIP-seq |
| ☒ | ☐ Flow cytometry |
| ☒ | ☐ MRI-based neuroimaging |

