## [Peer Review File · Nature Methods]

Peer Review Information

Manuscript Title: "ColabFold - Making protein folding accessible to all"

Corresponding author name(s): Sergey Ovchinnikov, Martin Steinegger, Milot Mirdita

Reviewer Comments & Decisions:

Decision Letter, initial version:
--

Dear Mr. Mirdita,

Your Brief Communication, "ColabFold - Making protein folding accessible to all", has now been seen by 2 reviewers. As you will see from their comments below, although the reviewers find your work of considerable potential interest, they have raised a number of concerns. We are interested in the possibility of publishing your paper in Nature Methods, but would like to consider your response to these concerns before we reach a final decision on publication.

We therefore invite you to revise your manuscript to address these concerns.

- * include a point-by-point response to the reviewers and to any editorial suggestions
- * please underline/highlight any additions to the text or areas with other significant changes to facilitate review of the revised manuscript
- * address the points listed described below to conform to our open science requirements

* ensure it complies with our general format requirements as set out in our guide to authors at www.nature.com/naturemethods

* resubmit all the necessary files electronically by using the link below to access your home page

[Redacted] This URL links to your confidential home page and associated information about manuscripts you may have submitted, or that you are reviewing for us. If you wish to forward this email to co-authors, please delete the link to your homepage.

We hope to receive your revised paper within 4 weeks. If you cannot send it within this time, please let us know. In this event, we will still be happy to reconsider your paper at a later date so long as nothing similar has been accepted for publication at Nature Methods or published elsewhere.

OPEN SCIENCE REQUIREMENTS

REPORTING SUMMARY AND EDITORIAL POLICY CHECKLISTS

DATA AVAILABILITY

We strongly encourage you to deposit all new data associated with the paper in a persistent repository where they can be freely and enduringly accessed. We recommend submitting the data to discipline-specific and community-recognized repositories; a list of repositories is provided here:

<http://www.nature.com/sdata/policies/repositories>

All novel DNA and RNA sequencing data, protein sequences, genetic polymorphisms, linked genotype and phenotype data, gene expression data, macromolecular structures, and proteomics data must be deposited in a publicly accessible database, and accession codes and associated hyperlinks must be provided in the “Data Availability” section.

Please include a “Data availability” subsection in the Online Methods. This section should inform readers about the availability of the data used to support the conclusions of your study, including accession codes to public repositories, references to source data that may be published alongside the paper, unique identifiers such as URLs to data repository entries, or data set DOIs, and any other statement about data availability. At a minimum, you should include the following statement: “The data that support the findings of this study are available from the corresponding author upon request”, describing which data is available upon request and mentioning any restrictions on availability. If DOIs are provided, please include these in the Reference list (authors, title, publisher (repository name),

identifier, year). For more guidance on how to write this section please see:
<http://www.nature.com/authors/policies/data/data-availability-statements-data-citations.pdf>

CODE AVAILABILITY

Please include a “Code Availability” subsection in the Online Methods which details how your custom code is made available. Only in rare cases (where code is not central to the main conclusions of the paper) is the statement “available upon request” allowed (and reasons should be specified).

For more information on our code sharing policy and requirements, please see:
<https://www.nature.com/nature-research/editorial-policies/reporting-standards#availability-of-computer-code>

MATERIALS AVAILABILITY

ORCID

Nature Methods is committed to improving transparency in authorship. As part of our efforts in this direction, we are now requesting that all authors identified as ‘corresponding author’ on published papers create and link their Open Researcher and Contributor Identifier (ORCID) with their account on the Manuscript Tracking System (MTS), prior to acceptance. This applies to primary research papers only. ORCID helps the scientific community achieve unambiguous attribution of all scholarly contributions. You can create and link your ORCID from the home page of the MTS by clicking on

'Modify my Springer Nature account'. For more information please visit www.springernature.com/orcid.

Sincerely,
Arunima

Arunima Singh, Ph.D.
Senior Editor
Nature Methods

Reviewers' Comments:

Reviewer #1:

Remarks to the Author:

This paper basically describes how programs like AlphaFold2 and RosettaFold can be run using the Google Colab service, and demonstrates that the results are at least comparable to those obtained by running the default implementations.

There really isn't a lot of new science to comment on here, as the results are mostly down to the AlphaFold2 codebase itself, which is already well documented in the literature. However, overall, I think ColabFold does make an important contribution, and allows relatively non-technical non-commercial users an easy way to run either AlphaFold or RosettaFold. The tool is nicely put together and relatively easy to use.

The article itself is reasonably clear, though I did find the methods section a bit dense and not an enjoyable read. I'm not sure much can be done about it, but there's a lot of quite disparate technical information dumped in there and really no real flow to it at all.

I do have a few specific issues to raise, however.

1. As far as I can see, the authors seem to completely ignore the fact that non-commercial users are not able to use the AF2 neural network weights. Unless the authors have permission from DeepMind to allow ColabFold users to ignore those license terms, the very title of this article is not really true. Protein folding is only accessible to academics, not “all”. Possibly the authors believe that it’s not their problem and that ColabFold just enables people to run the code IF they are licensed, but I think this would be a disingenuous argument. Really the authors should make this restriction very clear in the article, AND also very clear on the ColabFold page itself where the user first interacts with it. It could be argued that they are enabling people to ignore DeepMind’s licensing terms if they do not do this, and I note that DeepMind’s official colab implementation of AlphaFold does make those license terms clear, and so the authors really must address this. Maybe changing the title would be asking too much, but I do think a strong argument could be made that it’s not strictly accurate once this restriction is made clear, and I would certainly encourage the authors to come up with something more accurate.

2. The results in Fig.2a are generally as might be expected, but the obvious outliers are the few FM targets which one or other ColabFold run predicts very well, but the official AlphaFold2 implementations do not. I think the authors should comment on these cases. Are the alignments better or are there just more sequences picked up? It’s not even consistent between the two ColabFold runs. It really is somewhat confusing to me as a reader and I would like to hear more from the authors on this – perhaps with some supplementary data to back it up. On that note, I think it would be useful for the authors to make the input alignments used in Fig.2 – or at least the ones from their own method – available for other researchers to evaluate. Same goes for the 1762 alignments and models generated for M.jan.(see below).

3. I find the runtimes in Supplementary Fig.1 and the quoted ability to model 1762 proteins in 40h on one 3090 GPU hard to reconcile. That runtime is 1.4 minutes per structure. Impressive, but is this what a typical user would expect to get if they ran ColabFold on colab itself? The methods section describes hardware with 128 cores and 2Tb of RAM and an Ampere GPU, and then suggests that the 40h mentioned in the discussion section is just for the neural network inference. This is a ridiculous overselling of results. I’m not aware of any colab instance having access to 128 AMD cores and 2Tb of RAM, or an Ampere GPU, and even then, the 40h quoted ignores the time it took them to run the alignments even on their 128 core monster machine – albeit only one extra hour. I think the authors should produce more realistic timings given the aim is to make folding accessible to “all” – I don’t have access to any nodes with that spec, even on our multi-million dollar compute cluster. In fact, 128 cores and 2 Tb RAM (16 Gb per core) would be a reasonable spec for a whole small compute _cluster_. Using such a tailored system for benchmarking is just a synthetic tech demo. How long would it take to reproduce those results by a real user without access to such fancy hardware and under normal conditions of usage?

4. The article mentions from the off that ColabFold enables the use of both AlphaFold2 and RosettaFold. It's also the very last word of the concluding paragraph. Despite this, RosettaFold never seems to be discussed again in the paper itself. I think it's essential that RosettaFold results be included in Figure 2, and briefly discussed in the main text. It shouldn't be difficult or that time consuming for the authors to do this given the claims about speed - the data should be available within just a couple of hours, and this would be a great opportunity to give directly comparable results between the two methods to help guide users as to which to use i.e. where they are run on exactly the same targets and with the same alignments.

Small point on Figure 2b. The y-scale is not helpful here. It almost suggests that ColabFold can run a 460 residue protein in zero minutes. Log scale here please!

Reviewer #2:

Remarks to the Author:

The developments prompted by AlphaFold2 are only possible because of the availability of open-source implementations of AlphaFold2 (at least for inference.) Some of those developments, including the first use of AlphaFold2 for predicting multi-chain protein complexes, appeared in public discussion on online platforms—an excellent example of open and collaborative science. Much of those developments were made possible by ColabFold: a publicly available resource for the broad community of biologists (including those lacking access to computational resources), as it provides a fast and accurate pipeline for predicting protein structures for single chains, homo- and heteromer complexes, and it is based on AlphaFold2 (and RoseTTAFold). The core idea of ColabFold is to reduce the runtime for the MSAs search without compromising the system accuracy (as in the case of the AlphaFold-Colab). ColabFold is faster and more accurate than the original Jupyter Notebook for Google Colaboratory developed by the AlphaFold team! Thus, ColabFold is a fantastic publicly available resource for the broad community of biologists (including those without access to computational resources). Moreover, it may help trigger many future developments in structural biology and beyond.

Comments and suggestions:

The manuscript's introduction puts equal emphasis on AlphaFold2 and RoseTTAFold, whereas much of the paper's analysis is concerned with comparing ColabFold results (speed and accuracy) with AlphaFold2 (e.g., Fig.2, Supplementary Fig.1). I don't think RoseTTAFold methods are explicitly used in the paper (it is only mentioned in materials and methods.) This may be stated explicitly in the manuscript introduction for the benefit of clarity. On the other hand, the current wording of the introduction gives the impression (primarily for non-experts) that RoseTTAFold accuracy is on par with AlphaFold2 results, which is not the case.

-l 19: I suggest briefly explaining what those environmental databases are (mainly for the benefit of non-experts as they may not be familiar with BFD and MGnify).

- l 29: comma after AlphaFold2

-l 27: and (Supplementary Fig.1): it would be instructive to plot the runtime in the ColabFold pipeline (MSAs and model.) This will provide a vivid comparison between MSA generation in AlphaFold2 and ColabFold, as Fig.2 (b) reports overall inference time without distinguishing between model and MSAs.

-l 50: Specify that the 41 hours to compute a proteome (with protein < 1000 residues) was for the archaeon *Methanocaldococcus jannaschii* (see more below).

Fig.2 In Fig.2 (a), the magenta case should be called ColabFoldDB instead of ColabFold-ColabFoldDB.

-l 75: I wonder if the authors expect that re-training AlphaFold2 with ~30 sufficiently diverse sequences will lead to a model as accurate as of the original AlphaFold2.

-l 95: the paragraph starting at 95 suggests that in Fig.2 (a), the authors are comparing five methods, whereas, in the plot, they only report four methods. It seems that the ColabFold baseline is missing.

-l 110: The authors comment on the inability of ColabFold to accurately predict T1084. In addition, it would be interesting to discuss targets T1043, T1033, and T1064. First, ColabFold-BFD/MGnify is doing substantially better than the original AlphaFold2. On the other hand, although ColabFoldDB is achieving good performance on T1033 and T1064, it is performing poorly on T1043.

l- 118, discussion regarding predicting complexes and Fig.2c: the use of ColabFold for predicting complexes is very interesting (as it was already used to aid cryo-EM analysis), though the approach used (pairing procedure) is now superseded by AlphaFold-Multimer (Evans et al.)

Moreover, the results reported in Fig2 are very different in terms of their scope: Fig.2a for CASP targets domains benchmark ColabFold against AlphaFold2, whereas Fig.2c for protein complexes is doing an internal benchmarking between pairing modes, instead of comparing ColabFold analysis with other methods (ideally AlphaFold for protein complexes). Therefore, I think one should try to compare ColabFold for protein complexes with external methods and assess again the trade-off between speed (generating MSAs) and accuracy, or at least discuss such possibilities. Finally, I assume that the authors will include AlphaFold-Multimer in ColabFold.

I-149: the authors make the interesting observation that additional recycling iterations could improve the accuracy of predicted targets, and they mention the case of designed proteins without homologs and challenging targets. Supplementary Fig.7 only reports the results regarding designed proteins; it would be interesting to include an example of difficult targets and explain what difficult means in such context (FM category in CASP, shallow MSAs, etc.)

The performance of AlphaFold2 drops substantially for targets with MSAs < 30; I'm wondering if the authors considered analyzing such targets (for naturally occurring proteins) and if additional recycling iterations could help improve the model's performance in such case.

I- 164: the authors mention the prediction of the proteome of the archaeon *Methanocaldococcus jannaschii*, but no results are reported on the main text. I think the results in 'Proteome benchmark' (materials and methods), pLDDT, TM-score, etc., could be reported in the main text.

I-173 and final paragraph: as my earlier point, the paper's analysis concerns AlphaFold2 (and not RoseTTAFold).

Author Rebuttal to Initial comments

Dear Arunima,

We apologize for the late reply. Attached is our response letter and manuscript with highlighted changes in blue.

After we submitted this manuscript, Deepmind released the source code and model weights of AlphaFold-multimer as well as a Google Colaboratory Notebook allowing users to model complex structures. We have now fully integrated AlphaFold-multimer in ColabFold allowing users to model protein complexes using our fast MMseqs2 search server and compared ourselves to AlphaFold-multimer in a newly added benchmark.

Some other important changes that we included in this revised manuscript are the following:

- Described our command line tool in detail that allows for 90 times faster batch predictions compared to AlphaFold2 and include it into Figure 1
- Included ColabFold-RoseTTAFold to our CASP14 benchmark
- Improved text flow of the method section

We are confident that we addressed all of the remarks of the reviewers. Attached are the comments of the reviewers and our responses.

We would like to thank the reviewers for their feedback, resulting in a much improved manuscript in

our opinion.

Best regards,

Milot Mirdita, Sergey Ovchinnikov and Martin Steinegger

> Reviewer #1:

> Remarks to the Author:

> This paper basically describes how programs like AlphaFold2 and RosettaFold can be run
> using the Google Colab service, and demonstrates that the results are at least comparable
> to those obtained by running the default implementations. We would like to thank the reviewer
for their valuable feedback.

The manuscript describes not just how to run AlphaFold2/RoseTTAFold through Google Collaboratory
but

suggests multiple ways how to speed up these methods in general, such as by replacing HMMer/HHblits
with MMseqs2 as well as model optimizations. For this we developed a new search workflow and
developed modules within MMseqs2. Additionally we tweaked AlphaFold2 for batch computations by
avoiding recompiling its models as much as possible and by introducing an early stopping criterion.
Only due to this speed up is it now possible to predict a proteome in 48 hours, ~90 times faster
than the base AlphaFold2 system. Additionally, ColabFold was the first system that allowed
researchers to compute complexes using AlphaFold2. We show in the new Figure 2b that the
residue-index manipulation based complex modeling we implemented before AlphaFold-multimer was
released still occasionally exceeds the latter's performance.

We changed Figure 1 to include both interfaces to interact with ColabFold (web and
standalone command line). As well we added Figure 2d, which compares the runtime of

AlphaFold2 against our early stopping version of ColabFold-AlphaFold2's batch mode. To highlight
the command line application we added the following text to the main manuscript:

We hope this shows that ColabFold offers more than just a Google Collaboratory implementation of
AlphaFold2/RoseTTAFold.

> There really isn't a lot of new science to comment on here, as the results are mostly down
> to the AlphaFold2 codebase itself, which is already well documented in the literature.
> However, overall, I think ColabFold does make an important contribution, and allows
> relatively non-technical non-commercial users an easy way to run either AlphaFold or
> RosettaFold. The tool is nicely put together and relatively easy to use.

> The article itself is reasonably clear, though I did find the methods section a bit dense and

> not an enjoyable read. I'm not sure much can be done about it, but there's a lot of quite
> disparate technical information dumped in there and really no real flow to it at all.

We agree with the reviewer that our methods section is too dense. We added subheadings and short intros to the methods sections to improve the legibility of the methods text. We clustered the individual topics together into the following categories: "Executing ColabFold", "Replacing MSA generation in AlphaFold2/RoseTTAFold with MMseqs2", "ColabFold databases", "Modeling protein complexes with ColabFold", "Speeding up AlphaFold2's model inference", "Exposing advanced features" and "Benchmarking ColabFold".

> I do have a few specific issues to raise, however.

> 1. As far as I can see, the authors seem to completely ignore the fact that non-commercial
> users are not able to use the AF2 neural network weights. Unless the authors have
> permission from DeepMind to allow ColabFold users to ignore those license terms, the
> very title of this article is not really true. Protein folding is only accessible to academics,
not
> "all". Possibly the authors believe that it's not their problem and that ColabFold just
> enables people to run the code IF they are licensed, but I think this would be a
> disingenuous argument. Really the authors should make this restriction very clear in the
> article, AND also very clear on the ColabFold page itself where the user first interacts with
> it. It could be argued that they are enabling people to ignore DeepMind's licensing terms if
> they do not do this, and I note that DeepMind's official colab implementation of AlphaFold
> does make those license terms clear, and so the authors really must address this. Maybe
> changing the title would be asking too much, but I do think a strong argument could be
> made that it's not strictly accurate once this restriction is made clear, and I would certainly
> encourage the authors to come up with something more accurate.

We added additional clarifications to the license terms to each notebook. Recently, the AlphaFold team has dropped the non-commercial clause from the license terms, following that we also dropped the non-commercial clause from the databases we maintain (ColabfoldDB, PDB70). Therefore, ColabFold with AlphaFold2 is fully usable for "all" now. Therefore, we did not change the manuscript / title.

> 2. The results in Fig.2a are generally as might be expected, but the obvious outliers are
> the few FM targets which one or other ColabFold run predicts very well, but the official
> AlphaFold2 implementations do not. I think the authors should comment on these cases.
> Are the alignments better or are there just more sequences picked up? It's not even

> consistent between the two ColabFold runs. It really is somewhat confusing to me as a
 > reader and I would like to hear more from the authors on this – perhaps with some
 > supplementary data to back it up. On that note, I think it would be useful for the authors to
 > make the input alignments used in Fig.2 – or at least the ones from their own method –
 > available for other researchers to evaluate. Same goes for the 1762 alignments and
 > models generated for M.jan.(see below).

We uploaded all MSAs and predictions made by ColabFold and added these to the “Data availability” section.

Additionally, we added a Supplementary Table 1 and refer to it in the main manuscript. In this table, we describe the targets that most significantly differ between AlphaFold2 and ColabFold-AlphaFold2-BFD/MGnify.

> 3. I find the runtimes in Supplementary Fig.1 and the quoted ability to model 1762 proteins
 > in 40h on one 3090 GPU hard to reconcile. That runtime is 1.4 minutes per structure.
 > Impressive, but is this what a typical user would expect to get if they ran ColabFold on
 > colab itself? The methods section describes hardware with 128 cores and 2Tb of RAM and
 > an Ampere GPU, and then suggests that the 40h mentioned in the discussion section is
 > just for the neural network inference. This is a ridiculous overselling of results. I’m not
 > aware of any colab instance having access to 128 AMD cores and 2Tb of RAM, or an
 > Ampere GPU, and even then, the 40h quoted ignores the time it took them to run the
 > alignments even on their 128 core monster machine – albeit only one extra hour. I think
 > the authors should produce more realistic timings given the aim is to make folding
 > accessible to “all” – I don’t have access to any nodes with that spec, even on our
 > multi-million dollar compute
 > cluster. In fact, 128 cores and 2 Tb RAM (16 Gb per core) would be a reasonable spec for
 > a whole small compute _cluster_. Using such a tailored system for benchmarking is just a
 > synthetic tech demo. How long would it take to reproduce those results by a real user
 > without access to such fancy hardware and under normal conditions of usage?

We agree that the machine is oversized. We repeated the experiment on a machine with 24-cores and 512 GB RAM. The runtime for the search increased from 58 minutes to 113 minutes. We added additional explanation that this benchmark covers the command line tool ``colabfold_batch``, thus a proteome prediction should be repeatable by a user on their own cluster, or alternatively (if a sufficiently fast GPU is assigned to the user inside Google Colab and the user provides already computed MSAs) through the special purpose ColabFold batch notebook.

We added the following paragraph to the methods text and added a new Figure 2d showing the speed-ups achieved in the batch search mode:

With the colabfold_search wrapper to MMseqs2 we search against the ColabFoldDB ("ColabFoldDB") in 113 min on a system with an AMD EPYC 7402P 24-Core CPU (no hyperthreading) and 512GB RAM.

MMseqs2

had a maximum resident set size of 308 GB during the search.

> 4. The article mentions from the off that ColabFold enables the use of both AlphaFold2
> and RosettaFold. It's also the very last word of the concluding paragraph. Despite this,

> RosettaFold never seems to be discussed again in the paper itself. I think it's essential that
> RosettaFold results be included in Figure 2, and briefly discussed in the main text. It
> shouldn't be difficult or that time consuming for the authors to do this given the claims
> about speed - the data should be available within just a couple of hours, and this would be
> a great opportunity to give directly comparable results between the two methods to help
> guide users as to which to use i.e. where they are run on exactly the same targets and
> with the same alignments.

We extended the RoseTTAFold notebook to be easier to run on the user's own hardware and repeated the experiment with RoseTTAFold+PyRosetta. The benchmark results are now included in Figure 2a and in the speed-benchmark in Figure 2b.

> Small point on Figure 2b. The y-scale is not helpful here. It almost suggests that ColabFold
> can run a 460 residue protein in zero minutes. Log scale here please!

We updated Figure 2b to present the runtimes in log-scale. We show the targets on the x-axis as discrete values sorted by their length to improve legibility of the figure.

> Reviewer #2:

> Remarks to the Author:

> The developments prompted by AlphaFold2 are only possible because of the availability of
> open-source implementations of AlphaFold2 (at least for inference.) Some of those
> developments, including the first use of AlphaFold2 for predicting multi-chain protein
> complexes, appeared in public discussion on online platforms—an excellent example of
> open and collaborative science. Much of those developments were made possible by
> ColabFold: a publicly available resource for the broad community of biologists (including
> those lacking access to computational resources), as it provides a fast and accurate

- > pipeline for predicting protein structures for single chains, homo- and heteromer
- > complexes, and it is based on AlphaFold2 (and RoseTTAFold). The core idea of ColabFold
- > is to reduce the runtime for the MSAs search without compromising the system accuracy
- > (as in the case of the AlphaFold-Colab). ColabFold is faster and more accurate than the
- > original Jupyter Notebook for Google Colaboratory
- > developed by the AlphaFold team! Thus, ColabFold is a fantastic publicly available
- > resource for the broad community of biologists (including those without access to
- > computational resources). Moreover, it may help trigger many future developments in
- > structural biology and beyond.

We would like to thank the reviewer for the kind and encouraging words as well as the valuable comments.

> Comments and suggestions:

- > The manuscript's introduction puts equal emphasis on AlphaFold2 and RoseTTAFold,
- > whereas much of the paper's analysis is concerned with comparing ColabFold results
- > (speed and accuracy) with AlphaFold2 (e.g., Fig.2, Supplementary Fig.1). I don't think
- > RoseTTAFold methods are explicitly used in the paper (it is only mentioned in materials
- > and methods.) This may be stated explicitly in the manuscript introduction for the benefit of
- > clarity. On the other hand, the current wording of the introduction gives the impression
- > (primarily for non-experts) that RoseTTAFold accuracy is on par with AlphaFold2 results,
- > which is not the case.

- > l-173 and final paragraph: as my earlier point, the paper's analysis concerns AlphaFold2
- > (and not RoseTTAFold).

This is a very good point. We added RoseTTAFold to Figure 2 a and b. Here, we present RoseTTAFold's performance using the same input MSAs as the ones given to AlphaFold2. We have updated the main text and methods section accordingly.

- > -l 19: I suggest briefly explaining what those environmental databases are (mainly for the
- > benefit of non-experts as they may not be familiar with BFD and MGnify).

We added a paragraph to the main manuscript that reads:

These environmental databases contain billions of proteins extracted from metagenomic and -transcriptomic experiments, which often complement databases dominated by isolate genomes.

- > - l 29: comma after AlphaFold2 Thank you. We fixed the typo.

> -l 27: and (Supplementary Fig.1): it would be instructive to plot the runtime in the ColabFold pipeline (MSAs and model.) This will provide a vivid comparison between MSA generation in AlphaFold2 and ColabFold, as Fig.2 (b) reports overall inference time without distinguishing between model and MSAs.

We expanded Figure 2b to also include model inference times for AlphaFold2, ColabFold and RoseTTAFold. We therefore removed Supplementary Figure 1 since this information is redundant.

> Fig.2 In Fig.2 (a), the magenta case should be called ColabFoldDB instead of ColabFold-ColabFoldDB.

We updated the figures and the corresponding main/method texts to consistently refer to ColabFold and its models/databases.

> -l 75: I wonder if the authors expect that re-training AlphaFold2 with ~30 sufficiently diverse sequences will lead to a model as accurate as of the original AlphaFold2.

We do not think that AlphaFold2 could be retrained using only 30 diverse sequences but a single MSA with 30 diverse sequences can be enough to predict a high quality structure. We clarified this by directly referencing the alignment depth figure (5a) from the AlphaFold2 paper and changing the sentence to read:

“However, often an MSA with only a few (~30) sufficiently diverse sequences is enough to produce high quality predictions (see Jumper et al., Fig. 5a).”

The AlphaFold2 team has undertaken a substantial effort to produce extremely diverse MSAs for training (see “1.3 Self-distillation dataset” in the AlphaFold2 supplementary information).

> -l 95: the paragraph starting at 95 suggests that in Fig.2 (a), the authors are comparing five

> methods, whereas, in the plot, they only report four methods. It seems that the ColabFold baseline is missing.

We added RoseTTAfold to the list and refer to the individual runs as: ColabFold-RoseTTAfold-BFD/MGnify, ColabFold-AlphaFold2-BFD/MGnify and ColabFold-AlphaFold2-ColabFoldDB.

> -l 110: The authors comment on the inability of ColabFold to accurately predict T1084. In

- > addition, it would be interesting to discuss targets T1043, T1033, and T1064. First,
- > ColabFold-BFD/MGnify is doing substantially better than the original AlphaFold2. On the
- > other hand, although ColabFoldDB is achieving good performance on T1033 and T1064, it
- > is performing poorly on T1043.

We added a supplementary table, highlighting significant differences in prediction accuracy between the methods and refer to this in the main text with:

Supplementary Table 1 contains a list of further targets where ColabFold differed significantly from AlphaFold2.

- > I- 118, discussion regarding predicting complexes and Fig.2c: the use of ColabFold for
- > predicting complexes is very interesting (as it was already used to aid cryo-EM analysis),
- > though the approach used (pairing procedure) is now superseded byAlphaFold-Multimer
- > (Evans et al.)

- > Moreover, the results reported in Fig2 are very different in terms of their scope: Fig.2a for
- > CASP targets domains benchmark ColabFold against AlphaFold2, whereas Fig.2c for
- > protein complexes is doing an internal benchmarking between pairing modes, instead of
- > comparing ColabFold analysis with other methods (ideally AlphaFold for protein
- > complexes). Therefore, I think one should try to compare ColabFold for protein complexes
- > with external methods and assess again the trade-off between speed (generating MSAs)
- > and accuracy, or at least discuss such possibilities. Finally, I assume that the authors will
- > include AlphaFold-Multimer in ColabFold.

We agree. We implemented AlphaFold-multimer into ColabFold and compared its performance to the AlphaFold2 base system using the same benchmark applied in Evans et al. We present this benchmark in Figure 2c. This benchmark replaces the previous one.

- > I-149: the authors make the interesting observation that additional recycling iterations
- > could improve the accuracy of predicted targets, and they mention the case of designed
- > proteins without homologs and challenging targets. Supplementary Fig.7 only reports the
- > results regarding designed proteins; it would be interesting to include an example of
- > difficult targets and explain what difficult means in such context (FM category in CASP,
- > shallow MSAs, etc.)

- > The performance of AlphaFold2 drops substantially for targets with MSAs < 30; I'm
- > wondering if the authors considered analyzing such targets (for naturally occurring
- > proteins) and if additional recycling iterations could help improve the model's performance
- > in such case.

We reran ColabFold-AlphaFold2-BFD/MGnify using 3 and 12 recycle iterations, the average TM-score improved from 0.887 to 0.898. We added this benchmark as a new Supplementary

Figure 6. Especially targets with little MSA information profited from additional recycle iterations.

We added the following to the main manuscript:

Rerunning the CASP14 benchmark using 12 recycles resulted in an improvement of average TM-score from 0.887 to 0.898 (Supplementary Fig. 6). The largest improvement was in targets with little MSA information.

> -l 50: Specify that the 41 hours to compute a proteome (with protein < 1000 residues) was > for the archaeon *Methanocaldococcus jannaschii* (see more below).

> l- 164: the authors mention the prediction of the proteome of the archaeon > *Methanocaldococcus jannaschii*, but no results are reported on the main text. I think the > results in 'Proteome benchmark' (materials and methods), pLDDT, TM-score, etc., could > be reported in the main text.

We added the following text to the main text:

ColabFold's batch mode with early stopping can compute the proteome of *Methanocaldococcus jannaschii* in 48 h on a consumer GPU - a ~90 times speedup over AlphaFold2.

And

(5) We developed the command line tool `colabfold_batch` to predict structures on local machines. All together, we show that the proteome of 1762 proteins shorter than 1000 aa of *M. jannaschii* can be predicted in 48 h with early stopping at pLDDT of ≥ 85 on one Nvidia Titan RTX (Fig. 2d), while sacrificing little-or-no prediction accuracy (Methods "Proteome Benchmark"). The average pLDDTs of AlphaFold2 and ColabFold Stop ≥ 85 were 89.75 and 88.78 in a subsampled set of 50 proteins.

Decision Letter, first revision:

Our ref: NMETH-BC47477A

1st Mar 2022

Dear Dr. Mirdita,

Thank you for submitting your revised manuscript "ColabFold - Making protein folding accessible to all" (NMETH-BC47477A). It has now been seen by the original referees and their comments are below. The reviewers find that the paper has improved in revision, and therefore we'll be happy in principle to publish it in Nature Methods, pending minor revisions to comply with our editorial and formatting guidelines.

TRANSPARENT PEER REVIEW

Thank you again for your interest in Nature Methods Please do not hesitate to contact me if you have any questions.

Sincerely,
Arunima

Arunima Singh, Ph.D.
Senior Editor
Nature Methods

ORCID

IMPORTANT: Non-corresponding authors do not have to link their ORCIDs but are encouraged to do so. Please note that it will not be possible to add/modify ORCIDs at proof. Thus, please let your co-authors

know that if they wish to have their ORCID added to the paper they must follow the procedure described in the following link prior to acceptance:

Reviewer #1 (Remarks to the Author):

I thank the authors for addressing the points that I raised. I don't have any significant remaining concerns.

Reviewer #2 (Remarks to the Author):

The authors have addressed all my comments and I consider the manuscript suitable for publication.

Final Decision Letter:

11th Apr 2022

Dear Mr. Mirdita,

I am pleased to inform you that your Brief Communication, "ColabFold - Making protein folding accessible to all", has now been accepted for publication in Nature Methods. Your paper is tentatively scheduled for publication in our June print issue, and will be published online prior to that. The received and accepted dates will be October 29, 2021 and April 11, 2022. This note is intended to let you know what to expect from us over the next month or so, and to let you know where to address any further questions.

Your paper will now be copyedited to ensure that it conforms to Nature Methods style. Once proofs are generated, they will be sent to you electronically and you will be asked to send a corrected version within 24 hours. It is extremely important that you let us know now whether you will be difficult to contact over the next month. If this is the case, we ask that you send us the contact information (email, phone and fax) of someone who will be able to check the proofs and deal with any last-minute problems.

If, when you receive your proof, you cannot meet the deadline, please inform us at rjsproduction@springernature.com immediately.

Once your manuscript is typeset and you have completed the appropriate grant of rights, you will receive a link to your electronic proof via email with a request to make any corrections within 48 hours. If, when you receive your proof, you cannot meet this deadline, please inform us at rjsproduction@springernature.com immediately.

Once your paper has been scheduled for online publication, the Nature press office will be in touch to confirm the details.

Content is published online weekly on Mondays and Thursdays, and the embargo is set at 16:00 London time (GMT)/11:00 am US Eastern time (EST) on the day of publication. If you need to know the exact publication date or when the news embargo will be lifted, please contact our press office after you have submitted your proof corrections. Now is the time to inform your Public Relations or Press Office about your paper, as they might be interested in promoting its publication. This will allow them time to prepare an accurate and satisfactory press release. Include your manuscript tracking number NMETH-BC47477B and the name of the journal, which they will need when they contact our office.

About one week before your paper is published online, we shall be distributing a press release to news organizations worldwide, which may include details of your work. We are happy for your institution or funding agency to prepare its own press release, but it must mention the embargo date and Nature Methods. Our Press Office will contact you closer to the time of publication, but if you or your Press Office have any inquiries in the meantime, please contact press@nature.com.

If you are active on Twitter, please e-mail me your and your coauthors' Twitter handles so that we may tag you when the paper is published.

Please note that Nature Methods is a Transformative Journal (TJ). Authors may publish their research with us through the traditional subscription access route or make their paper immediately open access through payment of an article-processing charge (APC). Authors will not be required to make a final decision about access to their article until it has been accepted. Find out more about Transformative Journals

Authors may need to take specific actions to achieve compliance with funder and institutional open access mandates. If your research is supported by a funder that requires immediate open access (e.g. according to Plan S principles) then you should select the gold OA route, and we will direct you to the compliant route where possible. For authors selecting the subscription publication route, the journal's standard licensing terms will need to be accepted, including self-archiving policies. Those licensing terms

will supersede any other terms that the author or any third party may assert apply to any version of the manuscript.

To assist our authors in disseminating their research to the broader community, our SharedIt initiative provides you with a unique shareable link that will allow anyone (with or without a subscription) to read the published article. Recipients of the link with a subscription will also be able to download and print the PDF. As soon as your article is published, you will receive an automated email with your shareable link.

Please note that you and your coauthors may order reprints and single copies of the issue containing your article through Springer Nature Limited's reprint website, which is located at <http://www.nature.com/reprints/author-reprints.html>. If there are any questions about reprints please send an email to author-reprints@nature.com and someone will assist you.

Best regards,

Arunima

Arunima Singh, Ph.D.

Senior Editor

Nature Methods